# Contrasting Development of Canopy Structure and Primary Production in Planted and Naturally Regenerated Red Pine Forests

**Laura J. Hickey** [1,*], **Jeff Atkins** [1] , **Robert T. Fahey** [2], **Mark R. Kreider** [3], **Shea B. Wales** [1] **and Christopher M. Gough** [1]

[1]  Department of Biology, Virginia Commonwealth University, Richmond, VA 23220, USA
[2]  Department of Natural Resources and the Environment & Center for Environmental Sciences and Engineering, University of Connecticut, Storrs, CT 06269, USA
[3]  Wildland Resources Department, Utah State University, Logan, UT 84321, USA
[*]  Correspondence: hickeylj@vcu.edu; Tel.: 540-604-6041

**Abstract:** Globally, planted forests are rapidly replacing naturally regenerated stands but the implications for canopy structure, carbon (C) storage, and the linkages between the two are unclear. We investigated the successional dynamics, interlinkages and mechanistic relationships between wood net primary production ($NPP_w$) and canopy structure in planted and naturally regenerated red pine (*Pinus resinosa* Sol. ex Aiton) stands spanning ≥ 45 years of development. We focused our canopy structural analysis on leaf area index (LAI) and a spatially integrative, terrestrial LiDAR-based complexity measure, canopy rugosity, which is positively correlated with $NPP_w$ in several naturally regenerated forests, but which has not been investigated in planted stands. We estimated stand $NPP_w$ using a dendrochronological approach and examined whether canopy rugosity relates to light absorption and light–use efficiency. We found that canopy rugosity increased similarly with age in planted and naturally regenerated stands, despite differences in other structural features including LAI and stem density. However, the relationship between canopy rugosity and $NPP_w$ was negative in planted and not significant in naturally regenerated stands, indicating structural complexity is not a globally positive driver of $NPP_w$. Underlying the negative $NPP_w$-canopy rugosity relationship in planted stands was a corresponding decline in light-use efficiency, which peaked in the youngest, densely stocked stand with high LAI and low structural complexity. Even with significant differences in the developmental trajectories of canopy structure, $NPP_w$, and light use, planted and naturally regenerated stands stored similar amounts of C in wood over a 45-year period. We conclude that widespread increases in planted forests are likely to affect age-related patterns in canopy structure and $NPP_w$, but planted and naturally regenerated forests may function as comparable long-term C sinks via different structural and mechanistic pathways.

**Keywords:** *Pinus resinosa* Sol. Ex Aiton; red pine; net primary production; leaf area index; LiDAR; canopy rugosity; forest succession; light; fPAR; chronosequence; light use efficiency

---

## 1. Introduction

Forested landscapes are increasingly a mosaic of naturally regenerated and planted stands varying in age and structure. Globally, planted forests occupy 264 million ha and are increasing in area by an average of 2% annually, with planting expanding most rapidly in North America and Asia [1]. In most regions of the world, fast-growing conifers are preferentially planted, often supplanting naturally regenerated hardwoods and conifers, sometimes of the same species [2–4]. When compared

with their naturally regenerated counterparts containing the same species assemblages, planted forests have higher stem densities and are less structurally heterogeneous, containing trees more uniform in height, diameter, and spacing [5,6]. In addition to the direct effects of natural and planted regeneration pathways on structure, stand characteristics with known linkages to ecosystem functioning such as stem density, leaf area index (LAI), and structural heterogeneity change as forests age [7–10]. Though the effects of stand regeneration pathways and age on structure are well-characterized, the functional implications of a global rise in planted forests—particularly for *rates* of carbon (C) storage—are only minimally understood [4,11–14].

Forest structure is inherently coupled with rates of C storage in plant biomass, a large component of total net primary production (NPP) [15], suggesting the distinct structural profiles of natural and planted stands may precipitate divergent NPP trajectories. While the positive influence of leaf quantity and area (e.g., LAI) on forest production applies universally to naturally regenerated and planted forests, the effects of structural complexity—the degree of spatial heterogeneity in vegetation quantity and arrangement—on planted forest NPP are poorly understood [14]. Naturally regenerated forests containing more heterogeneously arranged vegetation may sustain higher rates of NPP [16–20], with the successional development of complex, multi-layered canopies [10] linked to improved light absorption and light-use efficiency [21]. Conversely, more structurally homogenous planted forests are engineered to rapidly produce a dense, concentrated layer of foliage following establishment, thereby maximizing light absorption. A legacy of structural uniformity at the time of planting may persist for decades [22], but self- and prescribed thinning, the formation of canopy gaps, and the ingrowth of unplanted vegetation may drive long-term increases or decreases in stand heterogeneity [13,23–25]. Whether stand structural complexity develops and affects NPP similarly regardless of the regeneration pathway is not known, but this knowledge is critical to forecasting how the rising global prominence of plantation forestry at the expense of naturally regenerated forests will alter the forest C sink.

We characterized the structural complexity, $NPP_w$, and canopy light absorption and light-use efficiency of naturally regenerated and planted red pine (*Pinus resinosa*) chronosequences spanning 50 years of stand development, with the principal goals of determining whether stands established via different regeneration pathways exhibit similar: 1) coupled canopy structure-$NPP_w$ patterns over time; and 2) light absorption and light-use efficiency relationships with structural complexity. Our study is motivated by recent evidence that novel and highly spatially integrated canopy structural complexity measures derived from light detection and ranging (lidar) technology strongly correlate with NPP in naturally regenerated forests [26], but that the functional significance and utility of these remotely sensed complexity measures have not been examined in planted forests. Our analysis focuses on the structural complexity measure "canopy rugosity", which summarizes the variance in vegetation density and distribution across horizontal and vertical canopy axes [26]. Increases in canopy rugosity are broadly linked with higher NPP in a variety of naturally regenerated forests [16,17,27]. We hypothesized that greater uniformity in the tree spacing, height, and diameter of planted stands would translate into lower canopy rugosity (i.e., structural complexity) relative to naturally regenerated stands, particularly in young forests, but that the effects of limited complexity on $NPP_w$ would be offset by high LAI. We anticipated that densely planted stands would absorb more light than naturally regenerated stands, but that the relatively low structural complexity of planted stands would constrain light-use efficiency.

## 2. Materials and Methods

### 2.1. Study Site Description

Our study was conducted at the University of Michigan Biological Station (UMBS), in northern lower Michigan (45.5594° N, 84.6738° W), where the mean annual air temperature is 5.5 °C and mean annual precipitation is 817 mm [27]. Our investigation centered on two red pine (*Pinus resinosa*)

dominated chronosequences: one established from natural regeneration (hereafter "natural" for brevity) and the other from planting (hereafter "planted"). The natural chronosequence included four stands ranging from 20 to 85 years old, established from wind-dispersed seed in 1932, 1947, 1973, and 1997. Planted chronosequence stands were initiated in 1948, 1953, 1958, 1965, and 1993 and were 25 to 70 years old, with all but the youngest stand displaying signs of one single row thinning at an unspecified time (Figure 1). Species diversity, as Shannon's Index (H), was < 1 in all stands, except for the oldest natural stand (Table 1). Prior to red pine, the landscape was comprised of secondary forest that regrew following the clear-cut harvesting of pre-settlement forests in the early 20th century [28]. Within each of the nine stands, we installed three circular 0.1-ha plots (*n* = 27 total plots). Acknowledging both the utility and limitations of non-replicated chronosequences [29], our study design sought to minimize edaphic and climatic variation among stands. All stands: were located on a common glacial outwash landscape and within 13 km of each other; had similar 50-year red pine site indexes (16.8 to 19.2 m); and were positioned primarily on Rubicon sand with some stands also containing the Blue Lake Loamy soil series (Table 1).

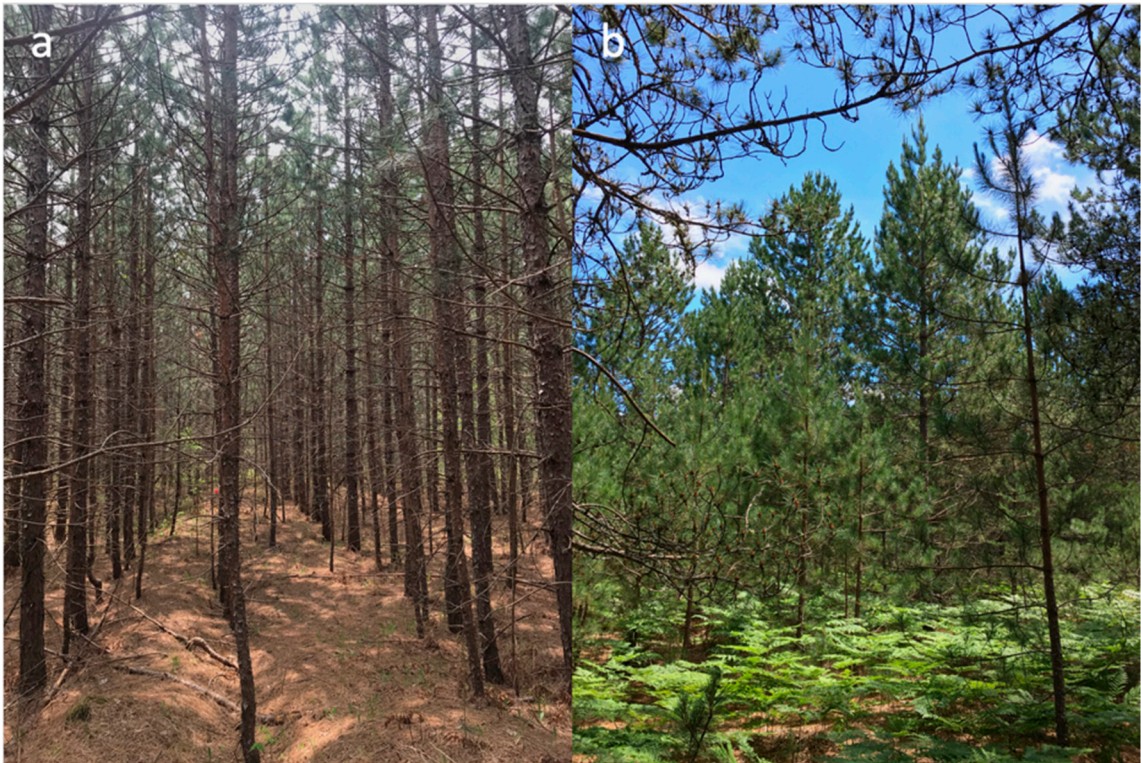

**Figure 1.** Planted 25-year-old (**a**) and naturally regenerated 20-year-old (**b**) red pine (*Pinus resinosa*) stands in north Michigan, USA. Planted stands are more structurally uniform, with relatively even stem spacing and heights, along with high stem densities. Naturally regenerated stands are uneven-aged, more sparsely vegetated, and contain trees varying in height.

*2.2. Canopy Structure*

We quantified canopy structure as rugosity (i.e., structural complexity) and LAI in natural and planted red pine stands using a Portable Canopy LiDAR (PCL) system. The PCL is an upward-facing, near-infrared, 2000-Hz pulsed laser that measures distance at sub-centimeter resolution. The PCL operator collects below-canopy data along a linear transect, which is used to generate a 2-dimensional 1 × 1 m voxelized "hit-grid" of vegetation density and distribution along vertical and horizontal axes [16,26,30]. Though several measures of canopy structure can be derived from vegetation hit-grids [26], we focus on canopy rugosity and leaf area index (LAI) because both are linked with light absorption and NPP [10,16,21]; however, canopy rugosity's relationship with light absorption

and production in planted forests is not known. Canopy rugosity, a measure of structural complexity, summarizes the degree of heterogeneity in vegetation distribution and density within a canopy [16]. PCL data were collected in each 0.1-ha plot along two 40 m transects running perpendicular along cardinal axes and intersecting at plot center [16]. Canopy rugosity and LAI were calculated using the open source *forestr* package [26] in R 3.5 (R Core Team, Vienna, Austria, 2019).

**Table 1.** Summary of stand characteristics by chronosequence. Species diversity was calculated as Shannon's Index of Diversity (H) and dominant species ranked according to percent basal area using canopy stem census data from 2017 and 2018. All stems with a diameter at breast height >5 cm were censused.

| Regeneration Pathway | Age (years) | Dominant Species | Species Diversity (H) | Soil Series | Site Index (m) | Basal Area ($m^2 ha^{-1}$) | Trees $ha^{-1}$ |
|---|---|---|---|---|---|---|---|
| Planted | | | | | | | |
| | 25 | *Pinus resinosa (97%)* *Prunus pensylvanica* L.f. *(3%)* | 0.08 | Rubicon sand | 16.8 | 39.5 | 3247 |
| | 53 | *Pinus resinosa (99%)* | 0.01 | Rubicon sand, Blue Lake loamy | 18.9 | 34.7 | 657 |
| | 60 | *Pinus resinosa (100%)* | 0.00 | Rubicon sand, Blue Lake loamy | 18.3 | 48.8 | 900 |
| | 65 | *Pinus resinosa (99%)* | 0.01 | Rubicon sand, Blue Lake loamy | 18.9 | 41.1 | 1036 |
| | 70 | *Pinus resinosa (94%)* *Quercus rubra* L. *(5%)* | 0.35 | Rubicon sand | 18.3 | 47.6 | 1163 |
| Natural | | | | | | | |
| | 20 | *Pinus resinosa (68%)* *Populus grandidentata (18%)* *Acer rubrum (4%)* | 0.69 | Rubicon sand | 16.8 | 7.9 | 524 |
| | 44 | *Pinus resinosa (95%)* | 0.58 | Rubicon sand | 18.9 | 22.1 | 404 |
| | 70 | *Pinus resinosa (77%)* *Quercus rubra* L. *(17%)* | 0.68 | Rubicon sand, Blue Lake loamy | 16.8 | 42.9 | 1400 |
| | 85 | *Pinus resinosa (59%)* *Populus grandidentata (19%)* *Acer rubrum (9%)* | 1.20 | Blue Lake loamy | 19.2 | 48.1 | 1247 |

## 2.3. Wood Net Primary Production

We estimated stand-level wood net primary production ($NPP_w$) from scaled dendrochronological measurements of annual stem wood increment. $NPP_w$ was quantified because it is estimated with relatively high certainty and closely parallels the net ecosystem carbon balance in forests (i.e., net ecosystem production) [31,32]. Following a plot-level census of all woody stems >8 cm in which diameter at breast height (DBH) and species were recorded, we selected, via random stratified sampling, 10 to 12 stems per plot from ranked species- and basal area-weighted distributions, coring no fewer than two stems per species within each plot ($n = 276$ total cores). The annual woody (i.e., xylem) growth increment of each core during the last five years, excluding the current year, was measured using a microscope and hand-operated stage. Cores unreadable under a scope were scanned using a Regent Instruments LA2400 Scanner and analyzed with WinDENDRO software. Woody stem growth increments measured using scope and scanner were highly correlated ($n = 31$, $p < 0.001$, $r^2 = 0.84$) with a slope of one, indicating the two approaches yield comparable values. We then estimated the 5-year wood C mass of each cored tree using species- and region-specific allometries [33] and carbon densities [31], reconstructing prior-year DBH by sequentially subtracting the annual increment from the measured reference diameter. We estimated the annual wood production of stems that were not cored as the product of the species- and plot-specific relative growth rates (i.e., individual stem wood production/wood mass) calculated from cored stems and applied to the DBH of the remaining censused tree population. Stand-scale $NPP_w$ was estimated as the sum of individual stem wood production averaged across plots and scaled to the hectare.

### 2.4. Light Absorption and Light-Use Efficiency

We estimated light absorption as the fraction of photosynthetically active radiation (fPAR) absorbed by the canopy. Plot-scale fPAR was measured using an AccuPAR LP-80 ceptometer along the two perpendicular 40-m transects used to sample canopy structure via a PCL. Specifically, above-canopy (i.e., open-sky) reference measurements were taken prior to below-canopy PAR observations 1 m above the forest floor at 1-m intervals along each transect. All measurements were taken within three hours of solar noon on cloudless days in July, 2018. Light-use efficiency was calculated as the quotient of $NPP_w$ and fPAR [34]. Stand-scale fPAR and light-use efficiency were estimated from plot averages.

### 2.5. Data Analysis

Our data and model analysis followed the precedent of prior studies [10,16,18] reporting linear and curvilinear relationships between stand age, $NPP_w$, canopy structure, and/or light absorption and light-use efficiency. We initially fit, tested the significance ($p < 0.05$) and ranked linear and 2-parameter curvilinear models separately for natural and planted forest stand data, treating the plot as the experimental unit. We selected the most parsimonious model by comparing adjusted $r^2$ values, applying a common model to natural and planted stands when the parameters from separate chronosequence-specific models had overlapping 95% confidence intervals (and thus were not significantly different from one another). We estimated the cumulative (45-year) production of natural and planted stands using an area-under-the-curve approach, comparing the two estimates statistically via a simple t-test. All data reported in figures is freely available numerically in published Supplementary Materials: DOI 10.6084/m9.figshare.832996.

## 3. Results

### 3.1. Age-Related Changes in NPP

Planted and naturally regenerated stands displayed different forest age-$NPP_w$ patterns (Figure 2). Planted stand $NPP_w$ was highest in the youngest, densest (25-year-old) stand, approaching 2500 kg C ha$^{-1}$ year$^{-1}$, and declined by half in >50-year-old stands. In contrast, the $NPP_w$ of natural pine stands was variable, from 950 kg C ha$^{-1}$ year$^{-1}$ to 1840 kg C ha$^{-1}$ year$^{-1}$, and did not display a significant pattern with age but did co-vary with site index (Table 1). Despite differences in age- $NPP_w$ relationships, planted and natural stands accumulated a similar amount of wood biomass over an overlapping 45-year period. The cumulative 25- to 70-year wood production values (Mg C ha$^{-1}$ ± 95% C.I.) of planted (72.1 ± 14.4) and natural (63.6 ± 14.5) pine stands were statistically indistinguishable ($p = 0.422$).

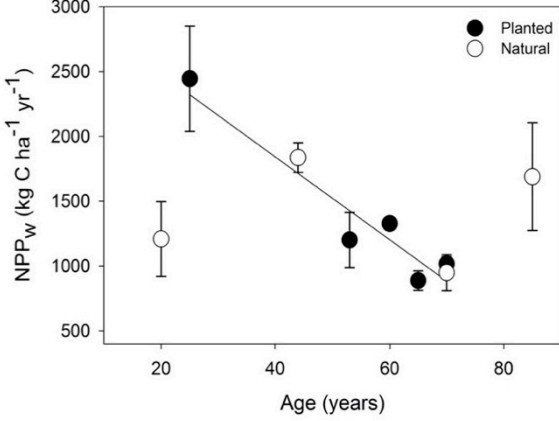

**Figure 2.** Mean annual wood net primary production ($NPP_w$; ± SE, $n = 3$) in relation to the stand age of planted pine and naturally regenerated red pine dominated stands. The solid regression line denotes a significant ($p < 0.05$) reduction in the planted pine with age ($r^2 = 0.65$).

## 3.2. Development of Canopy Structure

Trajectories of stand structural characteristics through stand development in planted and natural pine stands were opposite in sign for LAI but similar for canopy rugosity. Like $NPP_w$, LAI (Figure 3a) was highest in the youngest planted stand, peaking around 6, and declined thereafter with increasing age. The opposite was true for LAI in the natural pine stands, which steadily trended upward with age, plateauing at 6 in the oldest stands. In contrast, planted and natural pine stands shared a positive age-canopy rugosity relationship, with rugosity ranging from 1 m in the youngest stands to between 11 m and 15 m in the oldest planted and natural pine stands, respectively (Figure 3b). Together, these findings show that natural and planted pine stands developed structural complexity (i.e., canopy rugosity) similarly over time, despite displaying different successional patterns of LAI.

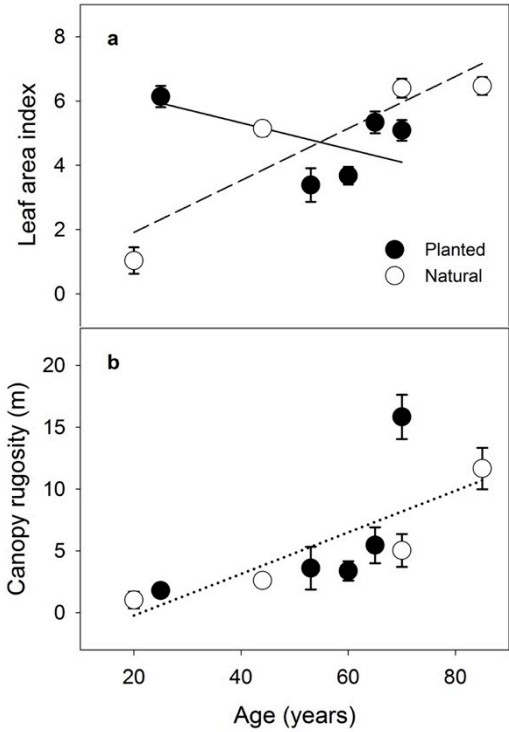

**Figure 3.** Mean leaf area index (**a**) and canopy rugosity (**b**) (± SE, *n* = 3) in relation to the age of planted and natural pine stands. Solid and dashed lines represent significant (*p* < 0.05) relationships with age in the planted and natural pine chronosequences, respectively (a, $r^2$ = 0.26 and 0.79), and a single dotted line represents a significant (*p* < 0.05) common relationship (b, $r^2$ = 0.47).

## 3.3. Structural Complexity–Production Relationships

Though structural complexity developed similarly with advancing age, planted and natural red pine stands had different canopy complexity-$NPP_w$ relationships (Figure 4). In planted red pine, we observed a negative non-linear relationship between canopy rugosity and $NPP_w$ that was driven by the youngest stand, which possessed low structural complexity but produced a large amount of wood biomass annually. Canopy rugosity spanned a similarly broad range in natural pine stands but was not correlated with $NPP_w$.

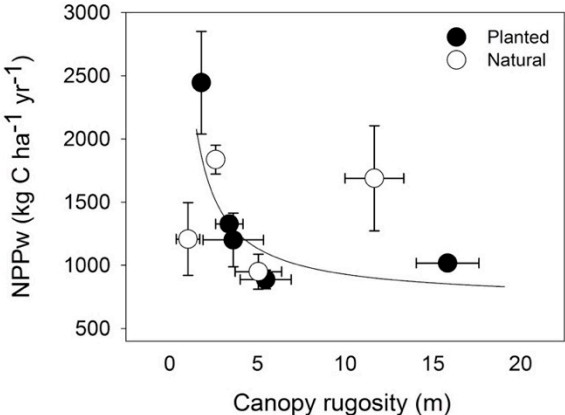

**Figure 4.** Stand mean annual wood net primary production (NPP$_w$, ± SE, *n* = 3) in relation to mean canopy rugosity (± SE, *n* = 3) in planted and natural pine stands. The solid line denotes a significant (*p* < 0.05) relationship in planted pine (*r*$^2$ = 0.43).

### 3.4. Structural Complexity–Light Relationships

We observed inconsistent and variable relationships between canopy rugosity and the light absorption and light-use efficiency of planted and natural red pine forests (Figure 5). Light-use efficiency in the planted pine stands declined sharply and non-linearly with increasing canopy rugosity, while natural stand light-use efficiency was not correlated with canopy rugosity (Figure 5a). In contrast, fPAR increased significantly with canopy rugosity in natural but not planted pine stands (Figure 5b).

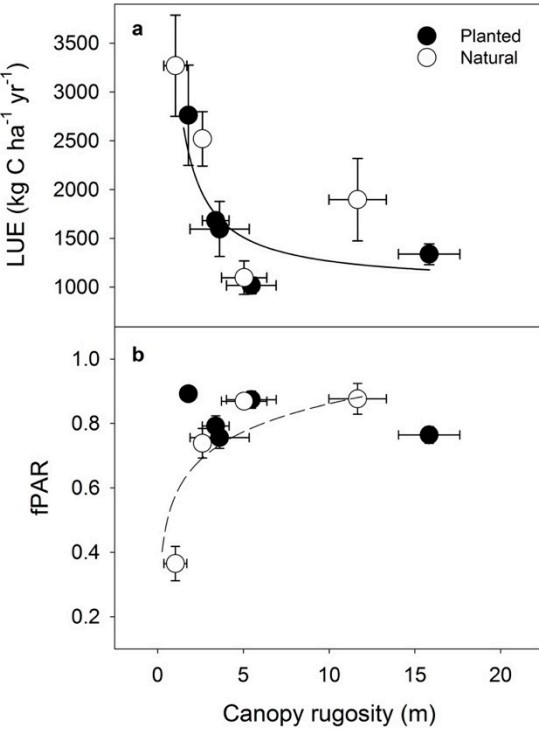

**Figure 5.** Mean light-use efficiency (LUE) (**a**) and absorbed fraction of photosynthetically active radiation (fPAR); (**b**) (± SE, *n* = 3) in relation to mean canopy rugosity (± SE, *n* = 3) in planted pine and natural pine stands. Solid and dashed lines represent significant (*p* < 0.05) change with age in the planted and natural chronosequences, respectively (a, *r*$^2$ = 0.46; b, *r*$^2$ = 0.49).

## 4. Discussion

Our results provide three primary advances toward understanding how different forest regeneration pathways affect structural complexity-$NPP_w$ relationships. First, and counter to our first hypothesis, we found that structural complexity, expressed in our study as canopy rugosity, develops similarly with age in red pine forests regardless of regeneration pathway, even when LAI trajectories diverge. Secondly, and partially supporting our first hypothesis, our observation that the $NPP_w$ of planted stands declined initially with increasing canopy rugosity indicates that canopy complexity— as we defined it—does not universally exert a positive influence over rates of wood production. Third, providing partial support for our second hypothesis, the relationships between canopy complexity and light absorption and light-use efficiency were different in planted and naturally regenerated red pine stands, but in unexpected ways. An additional synthetic finding that merits emphasis is the observation that, despite having different structural complexity-$NPP_w$ relationships, planted and natural red pine stands can accumulate similar quantities of C over a period of decades via different mechanisms.

Our finding that canopy rugosity–age relationships were similar in planted and naturally regenerated stands is surprising given the co-occurrence of opposite LAI-age patterns and prior results suggesting leaf quantity constrains canopy complexity. In accordance with other studies, stand heterogeneity and structural complexity features generally increase as forests age until a maximum level is reached [17,22,27,35,36], including in red pine ecosystems [24]. However, prior investigation of the structural complexity measure canopy rugosity in nearby naturally regenerated hardwood forests shows LAI and canopy rugosity moderately co-vary, suggesting the quantity of leaves available to construct a broad variety of canopy architectures may limit complexity [10,16,27]. Though our results from naturally regenerated red pine stands support a positive dependency of structural complexity on LAI, planted red pine structural complexity increased comparably with age, even as LAI declined, a result that suggests structural complexity developed via alternative means. As with any chronosequence-based study design [29], these relationships should be interpreted with proper caution, acknowledging that the stand age range sampled did not encompass very young stands (<20 years) and included age gaps between stands of >20 years.

Though the factors that led to a similar developmental trajectory of structural complexity in our planted and natural stands are not known, low-intensity and -frequency management activities—including thinning—may increase stand heterogeneity [13,37]. In addition, an increase in canopy complexity with age in the planted stands may coincide with understory establishment and reflect the increased complexity that the addition of a subcanopy can have in single-layered forests [38]. In multi-layered naturally regenerated forest canopies, age-related increases in structural complexity may emerge as outer canopy height increases and encompasses a broader range of tree heights and architectures. Despite likely differences in the developmental processes underlying structural complexity development in planted and naturally regenerated stands, theory and observations centered on the concept of resource complementarity suggest that canopies free of major disturbance organize similarly over time to optimize resource acquisition and resource-use efficiency [39–42].

While we anticipated a positive structural complexity-$NPP_w$ relationship in planted and naturally regenerated stands, negative and neutral relationships between stand structural complexity or heterogeneity and production are common. Focusing on canopy rugosity in similarly aged hardwood forests, we and others have observed a consistently positive relationship between structural complexity and $NPP_w$ [10,16,27]. However, the broader literature, which encompasses multiple biomes and complexity measures, suggests a more nuanced and variable relationship between structural complexity, broadly defined, and rates of biomass production. For example, a recent review by Ali [23] identified eight studies reporting positive, five citing negative, and two finding no effects of stand structural heterogeneity on annual wood biomass or volume production rates. The characterization of "heterogeneity" and "complexity" among the studies reviewed by Ali was variable, focusing on indexes that describe stem diameter similarity, and none of the studies surveyed used an approach similar to ours that integrates structural information across vertical and

horizontal axes of the canopy. The lack of consistency among studies in how structural complexity relates to production exposes a need for the standardization of theory, terminology, methodology to facilitate direct comparisons of structure-production relationships among sites and forest ecosystems.

In the planted red pine stands, a negative relationship between canopy rugosity and $NPP_w$, driven solely by the youngest and most productive stand, suggests a canopy structural property other than complexity may be the primary driver of high wood production in young planted stands. The very high stem density of the youngest planted stand (nearly 6x greater than that of the youngest naturally regenerated stand) supported high LAI, which, instead of canopy rugosity, may be the primary structural variable driving elevated levels of light absorption and light-use efficiency in young, planted forests [43]. Unlike naturally regenerated red pine stands, high tree density in the youngest planted stand appears to have facilitated high light absorption (as fPAR) and may have elevated light-use efficiency by allowing more complete and uniform access to light [14]. Numerous studies have investigated tree planting density and production, generally reporting higher NPP or wood volume production in more densely stocked young stands [44–49], and the high site utilization of high-density planting may override the positive effects of structural complexity in these forests.

In the broader context of $CO_2$ mitigation, our findings indicate that planted and natural stands can achieve similar long-term cumulative C storage in wood via separate structural development pathways. Though we found that planted red pine stands initially accumulated C rapidly in wood, naturally regenerated stands exhibited slightly lower, on average, but more stable NPP across ages, a pattern that was observed in other forest ecosystems [11,50,51] and which suggests a trade-off between $NPP_w$ stability and magnitude. While the application of our results to other forest ecosystems and C pools (e.g., leaves, soils) is not known, our results suggest that management approaches with different structural goals may result in similar long-term forest C storage or, more broadly, different stand structural pathways may lead to similar C storage outcomes. Whereas densely planted forests may utilize site resources thoroughly at a young age, more sparsely and gradually populated naturally regenerated stands may improve their site resource utilization over time as stem density, LAI, and structural complexity increase. As foresters increasingly weigh $CO_2$ mitigation options alongside conventional goals of timber production, additional investigation is needed to identify the variety of silvicultural-directed structural pathways that will lead to comparable long-term C storage outcomes.

## 5. Conclusions

Together, our findings indicate that planted and naturally regenerated red pine stands exhibit different canopy structural, including complexity, relationships with $NPP_w$, but, despite these differences, regeneration pathway has little effect on long-term wood C storage. From this core finding, we conclude that structural complexity may not exert the same positive influence over $NPP_w$ in planted and natural forests and, instead, other canopy structural features such as LAI may be more important drivers of wood production in planted forests. In practice, our results indicate that different regeneration approaches can achieve similar long-term C storage goals and, accordingly, forest managers possess a degree of flexibility when cultivating canopy structure for the purpose of maximizing carbon sequestration. Moving forward, we suggest the research community clarify and standardize definitions of stand "complexity" and "heterogeneity", and work toward greater understanding of what structural features and management approaches confer similar long-term functional outcomes.

**Supplementary Materials:** All data displayed in figures are freely available via Figshare, http://dx.doi.org/10.6084/m9.figshare.8329964.

**Author Contributions:** Conceptualization, L.J.H., J.A., M.R.K. and C.M.G.; methodology, L.J.H., J.A., M.K. and C.M.G.; formal analysis, L.J.H., J.A., M.R.K. and C.M.G.; investigation, L.J.H., J.A., M.R.K., S.B.W.; resources, C.M.G.; data curation, L.J.H.; writing—original draft preparation, L.J.H. and C.M.G.; writing—review and editing, R.T.F.; visualization, L.J.H.; supervision, J.A., R.T.F. and C.M.G.; project administration, J.A. and C.M.G.; funding acquisition, C.M.G.

**Funding:** This research was funded by the National Science Foundation (NSF) Division of Environmental Biology, grant number 1655095, and the NSF Division of Atmospheric and Geospace Sciences, grant number 1659338.

**Acknowledgments:** We thank the University of Michigan Biological Station (UMBS) for facilities support. L.J.H. is grateful for guidance from UMBS Data Manager, Jason Tallant; UMBS Research Experience for Undergraduate (REU) Directors, Dave Karowe and Steve Bertman; and, Virginia Commonwealth University Director of Undergraduate Research, Sarah Golding.

**Conflicts of Interest:** The authors declare no conflict of interest. The funders had no role in the design of the study; in the collection, analyses, or interpretation of data; in the writing of the manuscript, or in the decision to publish the results.

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
