# Peer review of "Contrasting Development of Canopy Structure and Primary Production in Planted and Naturally Regenerated Red Pine Forests"

_forests, doi:10.3390/f10070566_

Round 1
Reviewer 1 Report
I have few comments on this manuscript. The research presented in this manuscript makes a solid contribution to our understanding of how forest productivity relates to light use and structural complexity in planted and naturally regenerated forests. The manuscript is well written, with an implicit understanding by the authors of both the strength and weaknesses of the research. Aside from a few missing methodological details, my only substantive critique is that there needs to be a more straightforward acknowledgement in the Discussion about the degree to which the limited sample sizes could influence the interpretation of the data. For example, the age distribution of the planted stands is far from even, with one stand considerably younger than the others. While this is a reasonable limitation that presumably flows from the availability of suitable research sites, this should be acknowledged.
L52: The rate of C storage in plant biomass is similar, but not analogous to NPP. Please revise.
Table 1: Are the percentages in the “Dominant Species” column based on basal area? Density?
L131: Please specify that NPPw closely parallels net ecosystem C balance in forests.
L139: Please specify the sample size for this analysis.
L148-155: What time of year did this sampling occur? Three hours from solar noon is quite different in June than September.
L232: Change “mechanistic strategies” to “mechanisms” as the forests themselves do not strategize.
Reviewer 2 Report
General Comments:
The paper considers effect of regeneration method on structural complexity and its interlinkages to wood net primary production. Questions raised in the manuscript are very relevant and fit well to the scope of the Forests journal. Paper is simple, understandable with clear thought-line. On the other side, this work is based on a single, relatively small area and on only one tree species making it a case study that may not be best suited for the international journal. Partly it can be compensated by new conclusions of the work that are really interesting for forestry practice.
On a technical level the applied methods are well described and the discussion uses the full potential of the obtained results.
Specific Comments:
Although the article is written in a very simple and concise manner, it does not lack any important part or details.
